# Non-infectious diseases in infectious disease consultation: A descriptive study in a tertiary care teaching hospital

**Yoshiro Hadano** [1,2]*, **Takanori Matsumoto**[3]

**1** Division of Infection Control and Prevention, Shimane University Hospital, Izumo Shimane, Japan,
**2** Department of Infectious Diseases, St. Mary's Hospital, Kurume, Japan, **3** Department of Pharmacy,
St. Mary's Hospital, Kurume, Japan

* yhadano@med.shimane-u.ac.jp

infectious diseases in infectious disease
consultation: A descriptive study in a tertiary care
teaching hospital. PLoS ONE 18(12): e0295708.

University Hospital, TAIWAN

**Data Availability Statement:** All relevant data are
within the manuscript and its Supporting
Information files.

**Funding:** The authors received no specific funding
for this work.

## Abstract

In this retrospective study, we aimed to investigate the frequency, trend, and nature of non-infectious diseases (non-IDs) as the final diagnosis for patients during an infectious disease (ID) consultation in an acute care hospital in Japan. This study included adult inpatients who underwent ID consultations between October 2016 and March 2018. The demographic data, clinical manifestations, and final non-ID diagnoses of cases were explored. Among the 502 patients who underwent ID consultations, 45 (9.0%) were diagnosed with non-IDs. The most common diagnoses were tumors (22.2%, n = 10), connective tissue and collagen vascular diseases (13.3%, n = 6), other inflammatory diseases (8.9%, n = 4), and drug-induced fever (8.9%, n = 4). Multiple logistic regression analysis showed that the presence of consultations for diagnosis (odds ratio [OR], 22.0; 95% confidence interval [CI], 10.1–48.2; p<0.01), consultations from the internal medicine department (OR, 2.5; 95% CI, 1.2–5.2; p = 0.02), and non-bacteremia cases (OR, 5.2; 95% CI, 1.4–19.3; p = 0.01) were independently associated with diagnosed non-IDs. Non-IDs after ID consultations were mainly tumor-related, inflammatory diseases, and drug fever. The presence of consultations for diagnosis, consultations from the internal medicine department and non-bacteremia cases were related to non-IDs among ID consultations. Further research is needed to explore the frequency and pattern of non-IDs to improve the quality of ID consultations in daily practice.

## Introduction

Recently, infectious disease (ID) specialists have become important in medical care, and the value of ID consultations has been increasingly recognized in Japanese hospitals [1–4]. Diagnostic consultations in cases of unexplained fever or symptoms are challenging for ID consultants. Although the final diagnoses in many ID consultations are IDs, at times, the final diagnoses are non-infectious diseases (non-IDs), such as drug fever, collagen vascular disease, autoimmune disease, and cancer [5–9]. As a result, ID consultants must not only encounter IDs but also learn about non-IDs, especially in terms of diagnosis. A previous study showed six

**Competing interests:** The authors have declared that no competing interests exist.

(1.6%) inpatients received a new cancer diagnosis among the 380 inpatients who underwent ID consultations [8]. However, no epidemiologic studies have been conducted on the presentation of non-IDs during ID consultations. Thus, this study aimed to investigate the frequency, trend, and nature of non-IDs as the final diagnosis for patients during an ID consultation in an acute care hospital in Japan.

## Methods

### Study design

This single-center retrospective observational study was conducted at St. Mary's Hospital (a 1097-bed acute tertiary care teaching hospital in Kurume, Japan) between October 2016 and March 2018. The hospital does not have a diagnostic department, such as the Department of General Internal Medicine. The Department of Emergency Medicine is occasionally in charge of patients who either require critical care or have multisystem diseases. This study was approved by the Institutional Review Board (IRB) of St. Mary's Hospital (No. 17–0203) and conducted according to the principles of the Declaration of Helsinki. The requirement to obtain written informed consent from all participants was waived by the Institutional Review Board of St. Mary's Hospital because of the study's observational nature without any deviation from the current medical practice.

### Inclusion and exclusion criteria

We enrolled all consecutive inpatients aged ≥18 years for whom an ID consultation was requested during the study period. We analyzed the ID consultation database; patients who underwent formal ID consultations were eligible for the analysis, whereas those who underwent informal (so-called 'curbside') consultations, which imply consultations unrelated to patient management such as infection control and surgical antimicrobial prophylaxis, were excluded. The following demographic data were collected: age, sex, requesting department, consultation location (general ward or intensive care unit), bacteremia status, and non-ID diagnosis. The specialty of the requesting departments was categorized into the following two groups: internal and non-internal medicine. Internal medicine included the departments of cardiology, diabetes and endocrinology, gastroenterology, hematology, nephrology, neurology, respiratory, and rheumatology. Non-internal medicine included the departments of cardiovascular surgery, dermatology, emergency medicine, ear, nose, and throat, gynecology, neurosurgery, orthopedic surgery, psychiatry, pediatrics, plastic surgery, surgery, thoracic surgery, and urology. The initial reasons for the ID consultation were categorized as diagnosis and management (fever or elevated levels of inflammatory markers, including white blood cell count or C-reactive protein, fever of unknown origin (FUO), suspicion of infection, or positive blood culture) or treatment of established infections (management of already-diagnosed infections, such as intra-abdominal infection, respiratory infection, and urinary tract infection).

### Statistical analysis

Categorical variables were analyzed using either chi-square or Fisher's exact tests, while continuous variables were represented as medians with the interquartile range (IQR) and compared using the Mann–Whitney U test. Patients were allocated to the non-ID or ID group; a univariate logistic regression analysis was used to obtain the odds ratios (ORs) and 95% confidence intervals (CI) indicating the magnitude of the interrelationship among risk factors. Furthermore, we analyzed the differences in outcomes between diagnosed ID and non-ID using multivariable logistic regression (backward selection method), adjusting for possible

differences in background characteristics (sex and age). The Hosmer–Lemeshow goodness-of-fit test was used to assess model fit, with non-significant results considered adequate. Statistical significance was defined as two-sided p<0.05, and all statistical analyses were performed using the JMP Pro (version 13.0, SAS Institute, Cary, USA) and SAS software version 9.4 (SAS Institute, Cary, NC, USA).

## Results

Overall, 502 patients (≥18 years) underwent ID consultations during the study period. The baseline characteristics of the patients are presented in Table 1. The median age was 68 years (IQR, 56–79) (range 18–102 years). Among the 508 consultations, 195 (38.8%) cases were requested by the internal medicine department, while 307 (61.2%) were requested from staff not associated with the internal medicine department. The most frequently requesting consultations were from the surgery (n = 106, 21.1%), orthopedic surgery (n = 38, 7.6%), and emergency medicine and respiratory departments, respectively (n = 33, 6.6%) (Table 2).

In the final diagnoses, 457 (91.0%) and 45 (9.0%) patients were diagnosed with IDs and non-IDs, respectively. The clinical characteristics of the patients diagnosed with non-IDs are presented in Table 3. The most common diagnoses were tumors (22.2%, n = 10), connective tissue and collagen vascular diseases (13.3%, n = 6), other inflammatory diseases (8.9%, n = 4), and drug-induced fever (8.9%, n = 4). Other diseases, such as alcoholic liver cirrhosis or liver failure and hematoma, were also observed.

Seven and three cases of solid tumors and hematologic malignancy, respectively, were related to the tumor diagnoses. Connective tissue and collagen vascular diseases included adult-onset Still's disease (AOSD), antineutrophil cytoplasmic antibody (ANCA) associated

**Table 1. Characteristics of the infectious disease consultations (n = 502).**

| Variables | Number of patients |
|---|---|
| Age (years) (IQR) | 68 (56–79) |
| Sex | |
| Male | 280 (55.8%) |
| Female | 222 (44.2%) |
| Consultation places | |
| General wards (1069 beds) | 376 (74.9%) |
| Intensive care unit (28 beds) | 95 (18.9%) |
| Outpatients/emergency rooms | 31 (6.1%) |
| Consult reasons | |
| Diagnosis and management | 87 (17.3%) |
| Treatment of established infections | 415 (82.7%) |
| Specialty | |
| Internal medicine | 195 (38.8%) |
| (Cardiology, diabetes and endocrinology, gastroenterology, hematology, nephrology, neurology, respiratory, and rheumatology) | |
| Non-internal medicine | 307 (61.2%) |
| (Cardiovascular surgery, dermatology, emergency medicine, ENT, gynecology, neurosurgery, orthopedic surgery, psychiatry, pediatrics, plastic surgery, surgery, thoracic surgery, and urology) | |
| Bacteremia | 165 (32.9%) |

IQR, interquartile range; ENT, ear, nose, and throat

**Table 2. Characteristics of the infectious disease consultations.**

| Hospital departments | No. of patients |
|---|---|
| Surgery | 106 (21.1%) |
| Orthopedic surgery | 38 (7.6%) |
| Emergency medicine | 33 (6.6%) |
| Respiratory | 33 (6.6%) |
| Cardiovascular surgery | 32 (6.4%) |
| Plastic surgery | 31 (6.2%) |
| Gastroenterology | 29 (5.8%) |
| Nephrology | 28 (5.6%) |
| Neurology | 27 (5.4%) |
| Hematology | 25 (5.0%) |
| Dermatology | 23 (4.6%) |
| Gynecology | 20 (4.0%) |
| Neurosurgery | 16 (3.4%) |
| Diabetes and endocrinology | 14 (2.8%) |
| Cardiology | 14 (2.8%) |
| Rheumatology | 13 (2.6%) |
| Psychiatry | 11 (2.2%) |
| Ear, nose, and throat | 6 (1.2%) |
| Others | 2 (0.4%) |
| Total | 502 (100%) |

Others: thoracic surgery and urology

**Table 3. Characteristics of non-infectious diseases following the infectious disease consultations.**

| Disorders | Number of patients (%) |
|---|---|
| Tumor (n = 10 (22.2%)) | |
| Solid tumor (gallbladder cancer, colon cancer (n = 2), occult cancer, intrathecal metastasis of melanoma, renal metastasis of esophageal cancer, uterine cancer | 7 (15.6%) |
| hematologic malignancy (acute myelogenous leukemia, malignant lymphoma (n = 2)) | 3 (6.7%) |
| Connective tissue and collagen vascular diseases (n = 6 (13.3%)) | |
| Collagen vascular diseases (adult-onset Still's disease), Antineutrophil cytoplasmic antibody-associated vasculitis, systemic lupus erythematosus, and synovitis, acne, pustulosis, hyperostosis, osteitis) | 4 (8.9%) |
| Calcium pyrophosphate crystal deposition disease | 2 (4.4%) |
| Other inflammatory diseases (n = 4 (8.9%)) | |
| Inflammatory bowel disease (Crohn's disease and ulcerative colitis) | 2 (4.4%) |
| Kikuchi–Fujimoto diseases | 2 (4.4%) |
| Drug-induced fever | 4 (8.9%) |
| Others (n = 17 (37.8%)) | |
| Alcoholic liver cirrhosis or liver failure | 3 (6.7%) |
| Hematoma | 2 (4.4%) |
| Others (acyclovir encephalopathy, fracture of the ribs, Guillain-Barré syndrome, heat stroke, heart failure, ischemic colitis, monotonic syndrome, postoperative natural progression, rhabdomyolysis, unknown cause of erythema nodosum, urinary tract obstruction, and vitamin B1 deficiency) | 12 (28.9%) |
| Unknown | 5 (8.9%) |
| Total | 45 (100%) |

**Table 4. Infectious disease consultation characteristics.**

| Variables | Non-infectious diseases (n = 45) (%) | Infectious diseases (n = 457) (%) | *P*-value |
|---|---|---|---|
| Consultation places | | | |
| General wards | 37 (82.2) | 339 (74.2) | 0.37 |
| Intensive care unit | 5 (11.1) | 90 (19.7) | |
| Outpatients/emergency rooms | 3 (6.7) | 28 (6.1) | |
| Age (years) (IQR) | 68 (56–76) | 68 (56–79) | 0.56 |
| Sex (male) | 18 (40.0) | 204 (44.6) | 0.55 |
| Consultation request | | | |
| Diagnosis and management | 35 (77.8) | 52 (11.4) | <0.01 |
| Treatment of established infections | 10 (22.2) | 405 (88.6) | |
| Specialty | | | |
| Internal medicine | 27 (60.0) | 168 (36.8) | <0.01 |
| Non-internal medicine | 18 (40.0) | 289 (63.2) | |
| Bacteremia | 3 (6.7) | 162 (35.5) | <0.01 |

IQR, interquartile range

vasculitis, systemic lupus erythematosus (SLE), synovitis, acne, pustulosis, hyperostosis, osteitis, and calcium pyrophosphate crystal deposition (CPPD) disease. In addition to connective tissue and collagen vascular diseases, inflammatory bowel disease (Crohn's disease and ulcerative colitis) and Kikuchi–Fujimoto diseases were observed.

Significant differences were observed in the baseline characteristics between the two groups, including consultation request, specialty, and incidence of bacteremia. ID consultation for diagnosis and management was more common in the non-ID group (35 cases, 77.8% vs 52 cases, 11.4%, p<0.01), and the incidence of request for consultation by the internal medicine department was higher at the final diagnosis of IDs (27 cases, 60% vs 168 cases, 36.8%, p<0.01) (Table 4). Furthermore, multiple logistic regression analysis showed that the presence of consultations for diagnosis (OR, 22.0; 95% CI, 10.1–48.2; p<0.01), consultations from the internal medicine department (OR, 2.5; 95% CI, 1.2–5.2; p = 0.02), and non-bacteremia cases (OR, 5.2; 95% CI, 1.4–19.3; p = 0.01) were independently associated with non-IDs (Table 5). The goodness of fit tests in logistic regression were not statistically significant (Hosmer–Lemeshow test; p = 0.53).

## Discussion

The study identifies the characteristics of non-IDs diagnosed in ID consultations at a Japanese tertiary care hospital. Among the 502 patients who underwent ID consultations, 45 (9.0%) were diagnosed with non-IDs. Non-IDs identified following ID consultations were mainly tumor-related, inflammatory diseases, and drug fever. The presence of consultations for diagnosis, consultations from the internal medicine department and non-bacteremia cases were associated with non-IDs among ID consultations.

The details of cases diagnosed as non-ID following ID consultations have not been well understood previously. Regarding the frequency, a previous study of ID consultations at a cancer center in Japan found that 8.4% of cases were non-IDs in nature, which was similar to this study [4]. A study conducted at an Australian tertiary hospital revealed that 4.6% of cases were non-IDs [6]. These results suggest that non-infectious conditions account for approximately 5–10% of ID consultations.

The pattern of causes for FUO has shifted over time. In the early 1900s and mid-1900s, infectious causes, including tuberculosis and endocarditis, were more common. However,

**Table 5. Results of univariate and multivariate analyses to estimate the diagnosis of non-infectious diseases.**

| Variable | Unadjusted OR | | | Multivariate OR | | |
|---|---|---|---|---|---|---|
| | OR | 95% CI | p-value | aOR | 95% CI | p-value |
| Sex | | | | | | |
| Male | 1.0 (reference) | | | 1.0 (reference) | | |
| Female | 0.8 | 0.44–1.54 | 0.55 | 0.6 | 0.30–1.38 | 0.26 |
| Age[a] | 1.0 | 0.98–1.01 | 0.56 | 1.0 (reference) | 0.99–1.03 | 0.51 |
| Consultation request | | | | | | |
| Treatment of established infections | 1.0 (reference) | | | 1.0 (reference) | | |
| Diagnosis and management | 28.5 | 13.8–64.0 | <0.01 | 22.0 | 10.08–48.17 | <0.01 |
| Specialty | | | | | | |
| Non-internal medicine | 1.0 (reference) | | | 1.0 (reference) | | |
| Internal medicine | 2.6 | 1.38–4.82 | <0.01 | 2.5 | 1.18–5.20 | 0.02 |
| Bacteremia | | | | | | |
| Yes | 1.0 (reference) | | | 1.0 (reference) | | |
| No | 7.7 | 2.35–25.18 | <0.01 | 5.2 | 1.41–19.33 | 0.01 |

OR, odds ratio; aOR, adjusted odds ratio; CI, confidence interval

[a]Per 1 year increase

autoimmune and autoinflammatory conditions have recently become more prevalent causes of FUO [10–12]. Although epidemiology varies by country, recent studies in Japan have shown that autoimmune and inflammatory conditions are the most common causes of FUO [13, 14]. Therefore, it is important for ID physicians, including those in Japan, to be aware of the epidemiology and characteristics of common non-IDs, particularly fever-related illnesses.

Regarding tumors, there are cases not only of what is termed tumor fever but also those related to cancer diagnosis. The cases related to the diagnosis were solid tumors, particularly colon cancer and hematological malignancies. In a previous study, the frequency of establishing new diagnoses during ID consultation was low (1.6%) [8]. The most common malignancies of neoplastic fever are lymphoma, especially non-Hodgkin's, leukemia, renal cell carcinoma, hepatocellular carcinoma, or other tumors metastatic to the liver [12, 15]. Malignant lymphoma is a common differential diagnosis as a cause of fever or FUO [12, 16]; after the diagnostic workup with no evidence of IDs, based on the results of the additional consultation, bone marrow aspiration was recommended. Therefore, ID specialists should always follow the principle of 'tissue is the issue,' regardless of overconfidence bias, until proven otherwise [17].

Connective tissue and collagen vascular diseases are also major causes of non-IDs experienced during ID consultations [10–12]. In our study, there were cases of SLE, ANCA-related vasculitis, and AOSD. In fact, we have also encountered multiple cases of common diseases, such as CPPD [18]. Furthermore, we have received consultations on inflammatory bowel disease in fever cases. Although not investigated in this study, autoinflammatory disorders, such as familial Mediterranean fever, are also important [10, 11]. Regarding miscellaneous causes, drug-associated fever is a common cause of fever. The common cause of drug-associated fevers is antibiotics, such as beta-lactams [19]. Kikuchi disease is a benign condition of unknown cause usually characterized by cervical lymphadenopathy and fever [20]. This disease appears to be more common in Asia, particularly in Japan, and can occasionally present as an FUO [21].

ID physicians should broaden their knowledge of commonly encountered non-infectious conditions. As the final diagnoses could also include various non-IDs, ID physicians should

not only understand IDs but should also be familiar with the diagnosis of non-IDs that are commonly encountered. There is a need to improve clinical knowledge through continued medical education, case-based discussions, and lecture series in daily practice. Thus, each ID program should recognize opportunities for improvement based on their needs and conduct continuous training for ID fellows and specialists, particularly relating to the presentation of tumors, inflammatory diseases, and drug fever.

This study has some limitations related to its retrospective and uncontrolled nature. First, the sample size was relatively small due to the single-center design and the short duration of the study. Second, this retrospective study was conducted at a single center with no diagnostic department at an acute hospital in Japan. The results may not apply to other settings, including other countries with different consultation styles, such as university hospitals or hospitals with diagnostic departments, including a General Internal Medicine or Hospital Medicine Department. Finally, our results excluded informal consultations, which play an important role in the workload of ID physicians. Curbside consultations are an important issue, particularly when the number of IDs is low [22]. Nevertheless, our findings are valuable as they describe non-ID diagnoses following ID consultations in a tertiary care teaching hospital. However, further research is needed to elucidate the epidemiology of non-IDs for ID specialists.

In conclusion, we explored the frequency of non-IDs in patients after ID consultations. Common causes of non-IDs were tumors, inflammatory diseases, and drug fever. Further research is needed to examine the frequency and pattern of non-IDs to improve the quality of ID consultations in daily practice, prevent diagnostic errors, and improve patient prognosis.

## Supporting information

**S1 Data.**
(CSV)

## Acknowledgments

We thank all the clinical staff at St. Mary's Hospital for their dedicated patient care.
We would like to thank Editage for English language editing.

## Author Contributions

**Conceptualization:** Yoshiro Hadano, Takanori Matsumoto.

**Data curation:** Yoshiro Hadano.

**Formal analysis:** Yoshiro Hadano.

**Investigation:** Yoshiro Hadano, Takanori Matsumoto.

**Methodology:** Yoshiro Hadano.

**Project administration:** Takanori Matsumoto.

**Supervision:** Takanori Matsumoto.

**Validation:** Takanori Matsumoto.

**Visualization:** Takanori Matsumoto.

**Writing – original draft:** Yoshiro Hadano.

**Writing – review & editing:** Takanori Matsumoto.

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
