## [Decision Letter · Decision Letter 0]

3 Oct 2023

PONE-D-23-27435Non-infectious diseases in infectious disease consultation: A descriptive study in a tertiary care teaching hospitalPLOS ONE

Dear Dr. Hadano,

Thank you for submitting your manuscript to PLOS ONE. After careful consideration, we feel that it has merit but does not fully meet PLOS ONE’s publication criteria as it currently stands. Therefore, we invite you to submit a revised version of the manuscript that addresses the points raised during the review process.

We look forward to receiving your revised manuscript.

Kind regards,

Tai-Heng Chen, M.D.

Academic Editor

PLOS ONE

https://journals.lww.com/md-journal/Fulltext/2020/06190/Infectious_disease_consultations_and_newly.84.aspx

https://www.nature.com/articles/s41598-022-23392-6

3. In your revision ensure you cite all your sources (including your own works), and quote or rephrase any duplicated text outside the methods section. Further consideration is dependent on these concerns being addressed.

Reviewers' comments:

Reviewer's Responses to Questions

**Comments to the Author**

1. Is the manuscript technically sound, and do the data support the conclusions?

Reviewer #1: Yes

Reviewer #2: Yes

2. Has the statistical analysis been performed appropriately and rigorously? 

Reviewer #1: Yes

Reviewer #2: Yes

3. Have the authors made all data underlying the findings in their manuscript fully available?

Reviewer #1: No

Reviewer #2: No

4. Is the manuscript presented in an intelligible fashion and written in standard English?

Reviewer #1: Yes

Reviewer #2: Yes

5. Review Comments to the Author

Reviewer #1: Thank you for the opportunity to review this insightful paper.

This study investigates the non-infectious disease component of infectious disease consultations in acute care hospitals in Japan.

From the standpoint of enhancing the quality of ID consultations, this research is invaluable.

However, before publication, certain revisions are necessary.

Major comments:

1)The authors mention that the multiple logistic regression analysis revealed that "Diagnosis and management," "Internal medicine consultation," and "non-bacteremia cases" were independently associated with non-ID cases.

Yet, there is no subsequent discussion on these three outcomes.

It is imperative for the authors to address them.

2)While the authors elaborate on the distribution of non-ID cases over the period, there is a lack of detailed information regarding the ID cases.

If feasible, a comprehensive description of the ID cases would be beneficial.

Minor comments:

Table 3:

The figures in the category of connective tissue and collagen vascular diseases are inconsistent.

What does the ")" signify in "oesteitis)?" The ")" appears to be an careless mistake by the author.

Reviewer #2: The manuscript is a single-centre retrospective review of noninfectious diseases in patients who received an infectious diseases (ID) consult. The manuscript is clearly written, the conclusions are sound, and the main limitations are correctly summarised.

The article's main limitation is a lack of generalisability. The centre where patients were recruited lacks a general medical department. Thus the pattern of non-infectious diagnoses for which ID were consulted may be very different to that in a hospital where generalists (internists, etc.) are more commonplace, and who may be more comfortable making non-infectious diagnoses. The authors do acknowledge this in the limitations section, but it remains a significant limitation nonetheless.

I have only minor comments:

1. I wasn't clear in the methods section how patients with both an ID and a non-ID diagnosis were categorised. (E.g. a patient with Hodgkins Lymphoma and also a bacteraemia). Greater clarity on this point would be welcome, since it speaks to whether the ID consultation was appropriate or not.

2. In the abstract, lines 33-37 are largely redundant, since they restate the preceding sentences.

3. Findings such as ID consultation for "diagnosis and management [as opposed to for treatment of established infections] was more common in the non-ID group" are almost inevitable from the definitions used (if the ID problem was obvious, then there would be no reason to consult the ID division for diagnosis). This may be worth clarifying in the discussion.

6. PLOS authors have the option to publish the peer review history of their article (what does this mean?). If published, this will include your full peer review and any attached files.

Reviewer #1: No

Reviewer #2: No

---

## [Author Response · Author response to Decision Letter 0]

13 Oct 2023

5. Review Comments to the Author

We appreciate your valuable feedback, which will help us refine our manuscript and contribute to the overall quality of our research. Thank you for your time and consideration.

Reviewer #1: Thank you for the opportunity to review this insightful paper.

This study investigates the non-infectious disease component of infectious disease consultations in acute care hospitals in Japan.

From the standpoint of enhancing the quality of ID consultations, this research is invaluable.

However, before publication, certain revisions are necessary.

Major comments:

1)The authors mention that the multiple logistic regression analysis revealed that "Diagnosis and management," "Internal medicine consultation," and "non-bacteremia cases" were independently associated with non-ID cases.

Yet, there is no subsequent discussion on these three outcomes.

It is imperative for the authors to address them.

Thank you for your insightful comments. We have provided additional comment to the Discussion section (Discussion section, lines 187-196, page 14). This change has been made.

2)While the authors elaborate on the distribution of non-ID cases over the period, there is a lack of detailed information regarding the ID cases.

If feasible, a comprehensive description of the ID cases would be beneficial.

Thank you for your insightful comment. I have added the top 5 infectious diseases diagnosed at consultation to the Results section (lines 118-121, page 8).This change has been made.

Minor comments:

Table 3:

The figures in the category of connective tissue and collagen vascular diseases are inconsistent.

What does the ")" signify in "oesteitis)?" The ")" appears to be an careless mistake by the author.

Regarding the inconsistency in the figures for the category of connective tissue and collagen vascular diseases, we have thoroughly reviewed our data and made necessary corrections to ensure accuracy in the table.

Regarding the ")" in "oesteitis)," it appears to be a typographical error. We have corrected this appropriately to read "osteitis" . 

This change has been made.

Reviewer #2: The manuscript is a single-centre retrospective review of noninfectious diseases in patients who received an infectious diseases (ID) consult. The manuscript is clearly written, the conclusions are sound, and the main limitations are correctly summarised.

The article's main limitation is a lack of generalisability. The centre where patients were recruited lacks a general medical department. Thus the pattern of non-infectious diagnoses for which ID were consulted may be very different to that in a hospital where generalists (internists, etc.) are more commonplace, and who may be more comfortable making non-infectious diagnoses. The authors do acknowledge this in the limitations section, but it remains a significant limitation nonetheless.

We appreciate your thorough review of our manuscript and your acknowledgment of its clarity and sound conclusions. We also value your insight regarding the main limitation, which is the lack of generalizability.

To address this concern, we have added the following statement to the limitations section: "It is necessary to consider the possibility of conducting a multi-center study to improve the generalizability of the research."

I have only minor comments:

1. I wasn't clear in the methods section how patients with both an ID and a non-ID diagnosis were categorised. (E.g. a patient with Hodgkins Lymphoma and also a bacteraemia). Greater clarity on this point would be welcome, since it speaks to whether the ID consultation was appropriate or not.

Thank you for your valuable feedback. We initially categorized patients as either ID or non-ID based on the final diagnosis in the database. However, it has come to our attention that some cases involved a combination of both, and there were discrepancies in some of the data.

To address this issue, we have now divided patients into three groups: ID, non-ID, and those with a combination of both. We have also added a definition in the Methods section. Consequently, there have been modifications to both the aggregated results and the analysis results. Additionally, in the section related to tumors, we have provided a breakdown distinguishing between 1) Neoplastic fever caused by known tumors and 2) the diagnosis of new tumors.

2. In the abstract, lines 33-37 are largely redundant, since they restate the preceding sentences.

Thank you for your feedback. Lines 33-37 have been removed. This change has been made.

3. Findings such as ID consultation for "diagnosis and management [as opposed to for treatment of established infections] was more common in the non-ID group" are almost inevitable from the definitions used (if the ID problem was obvious, then there would be no reason to consult the ID division for diagnosis). This may be worth clarifying in the discussion.

Thank you for your feedback. I agree with the difficulty with the case definition. I have added an update to the discussion section.

---

## [Decision Letter · Decision Letter 1]

24 Nov 2023

Non-infectious diseases in infectious disease consultation: A descriptive study in a tertiary care teaching hospital

PONE-D-23-27435R1

Dear Dr. Hadano,

We’re pleased to inform you that your manuscript has been judged scientifically suitable for publication and will be formally accepted for publication once it meets all outstanding technical requirements.

Kind regards,

Tai-Heng Chen, M.D., Ph.D.

Academic Editor

PLOS ONE

Reviewers' comments:

Reviewer's Responses to Questions

**Comments to the Author**

1. If the authors have adequately addressed your comments raised in a previous round of review and you feel that this manuscript is now acceptable for publication, you may indicate that here to bypass the “Comments to the Author” section, enter your conflict of interest statement in the “Confidential to Editor” section, and submit your "Accept" recommendation.

Reviewer #1: All comments have been addressed

Reviewer #2: All comments have been addressed

2. Is the manuscript technically sound, and do the data support the conclusions?

Reviewer #1: Yes

Reviewer #2: Yes

3. Has the statistical analysis been performed appropriately and rigorously? 

Reviewer #1: Yes

Reviewer #2: Yes

4. Have the authors made all data underlying the findings in their manuscript fully available?

Reviewer #1: No

Reviewer #2: Yes

5. Is the manuscript presented in an intelligible fashion and written in standard English?

Reviewer #1: Yes

Reviewer #2: Yes

6. Review Comments to the Author

Reviewer #1: (No Response)

Reviewer #2: Thank you - the authors have addressed my comments to my satisfaction. This has entailed a significant reworking of their primary data, but this is for the best.

7. PLOS authors have the option to publish the peer review history of their article (what does this mean?). If published, this will include your full peer review and any attached files.

Reviewer #1: No

Reviewer #2: No

---

## [Editor Report · Acceptance letter]

30 Nov 2023

PONE-D-23-27435R1 

Non-infectious diseases in infectious disease consultation: A descriptive study in a tertiary care teaching hospital 

Dear Dr. Hadano:

I'm pleased to inform you that your manuscript has been deemed suitable for publication in PLOS ONE. Congratulations! Your manuscript is now with our production department. 

Kind regards, 

on behalf of

Dr. Tai-Heng Chen 

Academic Editor

PLOS ONE